# In vivo reprogramming of pancreatic acinar cells to three islet endocrine subtypes

Weida Li[1†], Mio Nakanishi[1,2†], Adrian Zumsteg[1], Matthew Shear[1], Christopher Wright[3], Douglas A Melton[1], Qiao Zhou[1*]

[1]Department of Stem Cell and Regenerative Biology, Harvard University, Cambridge, United States; [2]Stem Cell and Cancer Research Institute, McMaster University, Ontario, Canada; [3]Department of Cell and Developmental Biology, Vanderbilt University School of Medicine, Nashville, United States

**Abstract** Direct lineage conversion of adult cells is a promising approach for regenerative medicine. A major challenge of lineage conversion is to generate specific cell subtypes. The pancreatic islets contain three major hormone-secreting endocrine subtypes: insulin[+] β-cells, glucagon[+] α-cells, and somatostatin[+] δ-cells. We previously reported that a combination of three transcription factors, Ngn3, Mafa, and Pdx1, directly reprograms pancreatic acinar cells to β-cells. We now show that acinar cells can be converted to δ-like and α-like cells by Ngn3 and Ngn3+Mafa respectively. Thus, three major islet endocrine subtypes can be derived by acinar reprogramming. Ngn3 promotes establishment of a generic endocrine state in acinar cells, and also promotes δ-specification in the absence of other factors. δ-specification is in turn suppressed by Mafa and Pdx1 during α- and β-cell induction. These studies identify a set of defined factors whose combinatorial actions reprogram acinar cells to distinct islet endocrine subtypes in vivo.

*For correspondence: qiao_zhou@harvard.edu

†These authors contributed equally to this work

Competing interests: The authors declare that no competing interests exist.

## Introduction

Cellular reprogramming is a rapidly expanding area of regenerative medicine. With suitable reprogramming factors, adult cells can be instructively converted to induced pluripotent stem cells (pluripotent reprogramming) or other types of adult cells (lineage reprogramming) (*Gurdon and Melton, 2008*; *Graf and Enver, 2009*). Induced pluripotent stem cells (iPS) can be differentiated into many cell types in the body. However, the generation of iPS cells and their subsequent differentiation is a lengthy and technically demanding process. Lineage conversion between adult cell types offers a promising alternative, directly producing defined cell types in vitro or even in vivo that may be used for disease modeling and cellular therapies (*Zhou and Melton, 2008*; *Vierbuchen and Wernig, 2011*). Recent examples of lineage reprogramming include the conversion of pre-B cells to macrophages, pancreatic acinar, α-cells, and gut cells to insulin-secreting β-cells, cardiac fibroblasts to cardiomyocyte-like cells, amniotic cells to endothelial cells, and skin fibroblasts to neurons, oligodendrocytes, neural precursors, or blood progenitors (*Xie et al., 2004*; *Zhou et al., 2008*; *Ieda et al., 2010*; *Szabo et al., 2010*; *Thorel et al., 2010*; *Vierbuchen et al., 2010*; *Caiazzo et al., 2011*; *Yang et al., 2011*; *Ginsberg et al., 2012*; *Han et al., 2012*; *Song et al., 2012*; *Talchai et al., 2012*; *Thier et al., 2012*; *Najm et al., 2013*; *Yang et al., 2013*).

Despite increasing success of lineage conversion, a major challenge of this approach is to direct the formation of specific cell types: there is a great diversity of cell types in the adult body, and many of them are further differentiated into closely related subtypes. The most extensive subtype diversification can be found in the mammalian central nervous system, where hundreds of neuronal subtypes

**eLife digest** In mammals, the pancreas is responsible for controlling blood sugar by secreting insulin from specialized β-cells. Other cells in the pancreas, called δ-cells and α-cells, secrete other hormones to assist the β-cells. Diabetes is caused when this system breaks down: either the body attacks its own β-cells (type I diabetes), or the body stops responding properly to insulin (type II).

Type I diabetes is usually treated with insulin injections, but there is increasing interest in the possibility of replacing the defective β-cells instead. Building on previous work in which a fourth type of pancreatic cell, called an acinar cell, was reprogrammed to become a β-cell, Li et al. have now shown that the same technique can be used to produce α- and δ-cells as well. Just as the reprogrammed β-cells secreted insulin, like real β-cells, the reprogrammed α- and δ-cells also behaved like real α- and δ-cells.

The reprogramming technique relies on using a combination of three transcription factors—which are called Ngn3, Pdx1 and Mafa—to treat the acinar cells from mice. Previously, it was shown that using a combination of all three transcription factors reprogrammed the acinar cells to become β-cells. Now, Li et al. show that the Ngn3 transcription factor on its own appears to suppress certain genes that are usually expressed in acinar cells, and goes on to cause the acinar cells to become δ-cells. However, a combination of Ngn3 and Mafa produces a mixture of α- and δ-cells. The next challenge is to adapt this reprogramming technique to generate different types of hormone secreting cells from human tissue sources in order to explore its therapeutic potential.

exist. A few mammalian neuronal subtypes, including dopaminergic-like and motoneuron-like cells, have been produced from fibroblast conversion (*Caiazzo et al., 2011*; *Son et al., 2011*); methods to generate many others remain to be defined. To study subtype specification in lineage reprogramming, it is necessary to first establish models, where a defined set of factors promote formation of distinct subtypes. A recent study in *Caenorhabditis elegans* provided such an example, where removal of a chromatin factor confers neurogenic competence to germ cells, which can be subsequently converted to different neuronal subtypes by neuron selector genes (*Tursun et al., 2011*).

To establish models of mammalian subtype specification in lineage reprogramming, we focused our studies in a relatively simple system, the adult pancreas, where the endocrine islets are surrounded by acinar cells, a type of exocrine cells that secret digestive enzymes. The islets contain three major endocrine subtypes: insulin[+] β-cells, glucagon[+] α-cells, and somatostatin[+] δ-cells. β-cells secret insulin and play a key role in blood glucose regulation, whereas α- and δ-cells secrete glucagon and somatostatin to support β-cell function (*Edlund, 2001*; *Jensen, 2004*).

We reported previously that pancreatic acinar cells can be directly converted to insulin[+] β-cells in adult mouse pancreas by combined actions of three transcription factors, Ngn3, Pdx1, and Mafa (referred to as M3 factors) (*Zhou et al., 2008*). We now report that acinar cells can also be converted to the other endocrine subtypes, namely, somatostatin[+] δ-like cells and glucagon[+] α-like cells, by Ngn3 and Ngn3+Mafa respectively. A defined set of factors can therefore reprogram acinar cells to the three major islet endocrine subtypes. Further studies indicate that Ngn3, but not Mafa and Pdx1, promotes establishment of a generic endocrine state in acinar cells at the onset of reprogramming by suppressing acinar fate-regulators and activating pan-endocrine genes. Ngn3 also promotes δ-subtype specification in the absence of other factors. Mafa and Ngn3 in turn suppress δ-specification in α- and β-cell formation, thus ensuring creation of singular endocrine subtypes. Our studies establish a series of models where combinatorial functions of defined factors convert pancreatic acinar cells to three distinct endocrine subtypes in vivo. These models provide a powerful system to gain mechanistic understanding of the lineage reprogramming process.

## Results

### Reprogramming acinar to δ-, α-, and β-like endocrine cells

We have previously reported that pancreatic acinar cells can be converted to insulin[+] β-like cells by the combined activity of three reprogramming factors: Ngn3, Mafa, and Pdx1, referred to as M3 factors (*Zhou et al., 2008*). Employing the same experimental system of adenoviral expression in adult mouse

pancreas, which specifically targets acinar cells (*Figure 1A*, *Figure 1—figure supplement 1*), we examined the role of individual M3 factors in endocrine reprogramming. Surprisingly, Ngn3 alone induced formation of somatostatin+ (Sst) cells in approximately 40% of infected cells (*Figure 1B–D*), whereas Mafa or Pdx1 alone did not induce any hormone positive cells (*Figure 1—figure supplement 2*). In addition, co-infection of Ngn3- and Mafa-induced formation of both glucagon+ (Gcg) and somatostatin+ cells, which are distinct from each other (*Figures 1E,F*). The other two-factor combinations, Ngn3 with Pdx1 and Pdx1 with Mafa, did not yield hormone positive cells (*Figure 1—figure supplement 2*). Somatostatin and glucagon are the principle hormones of endocrine δ- and α-cells. These data suggest that different combinations of three reprogramming factors could convert pancreatic acinar cells in vivo to the three major islet endocrine cell types: δ-, α- and β-cells. The expression of reprogramming factors in δ- and α-cell induction is transient (*Figure 1—figure supplement 3*), similar to β-cell induction using the same experimental approach (*Zhou et al., 2008*). To confirm the identity of the induced Sst+ and Gcg+ cells, we examined whether the induced cells have key features of endogenous δ- and α-cells.

## Ngn3 converts acinar to δ-like cells

Among the major islet endocrine cell types, relatively little is known about δ-cell biology and genes important for δ-cell development and function. Among the few δ-cell-specific genes identified are *somatostatin* and *cholecystokinin receptor B (Cckbr)* (*Morisset et al., 2000*). Our analysis revealed that the majority of induced δ-like cells co-express Sst and Cckbr 30 days after induction (87 ±7% by immunohistochemistry, *Figure 2A*). The Sst+ cells also express the endocrine factors Pax6 and synaptophysin (*Figure 2B,C*). The Sst+ induced δ-cells were present in adult pancreas 2 months after induction (*Figure 2—figure supplement 1*).

A major method to recognize and distinguish the different islet endocrine subtypes is by ultrastructural analysis. In particular, the secretory granules of each islet subtype have characteristic morphology (*Larsson et al., 1976*; *Leiter et al., 1979*). Electron microscopy analysis revealed that the secretory granules of induced δ-cells are spherical or ellipsoidal with matrix filling the entire granule space (*Figure 2E,E'*). This morphology is typical of endogenous δ-cells (*Figure 2D,D'*) and distinct from that of α-or β-granules (*Figure 2—figure supplement 2*). In addition, we observed that the induced δ-cells were embedded among acinar cells (*Figure 2E*, arrow points to dense assembly of endoplasmic reticulum in a neighboring acinar cell), consistent with their origins from acinar cells. In contrast, endogenous δ-cells reside exclusively within islets (*Figure 2D*).

To further characterize the induced δ-cells, we generated gene expression data from FACS purified induced δ-cells (day 10 after infection) using mCherry, a fluorescent marker coexpressed with Ngn3 (*Figure 1A*). Cherry+ cells contain approximately 40% induced δ-cells. Gene profiling with illumina arrays yielded 1283 genes enriched in the induced δ-cells (30 days after induction) relative to acinar cells (GEO: GSE52522). Because there is currently no method available that allows purification of endogenous δ-cells, we compared their expression profile with that of whole islets, which are comprised largely of β-cells. 632 of the induced genes (49%) overlapped with the islet-enriched gene set that we previously reported (*Figure 2F*) (*Zhou et al., 2008*) (GEO: GSE12025). Given that δ-cells represent a minor fraction of total mouse islet cells (5.5% as determined by FACS, see *Figure 2—figure supplement 4*), the non-overlapping genes (651) may contain δ-cell-enriched factors that are underrepresented in the whole-islet samples. Indeed, among the top 30 most highly induced genes in the δ-like cells, many show low expression in whole islet samples (*Figure 2G*). We detected up-regulation of *Hhex* (*Figure 2G*), a gene recently implicated in δ-cell biology (72nd ADA abstract, Klaus Kaestner lab). In contrast, glucagon, insulin, and the β-cell marker Nkx6.1 are abundantly expressed in islets but absent from induced δ-cells (*Figure 2G*).

We analyzed the DNA methylation status of several gene promoters to assess epigenetic changes in the conversion of acinar to δ-like cells. These genes included *Somatostatin* (δ-cells), *Amylase2* (acinar cells), and *Insulin2* (β-cells). Studies have shown that insulin2 gene expression is regulated by DNA sequences located within approximately 400 bp upstream of the transcription start site (TSS) (*Hay and Docherty, 2006*). Similarly, approximately 200 bp of the *Amylase 2* promoter is sufficient to direct acinar-specific expression (*Minami et al., 2005*). We therefore assayed CpGs located in these critical promoter regions (*Figure 2—figure supplement 3*). There are very limited studies on the *Somatostatin* promoter so we assayed seven CpGs that fall within a highly conversed promoter region (*Figure 2—figure supplement 5*). We adapted an intracellular FACS protocol to purify endogenous

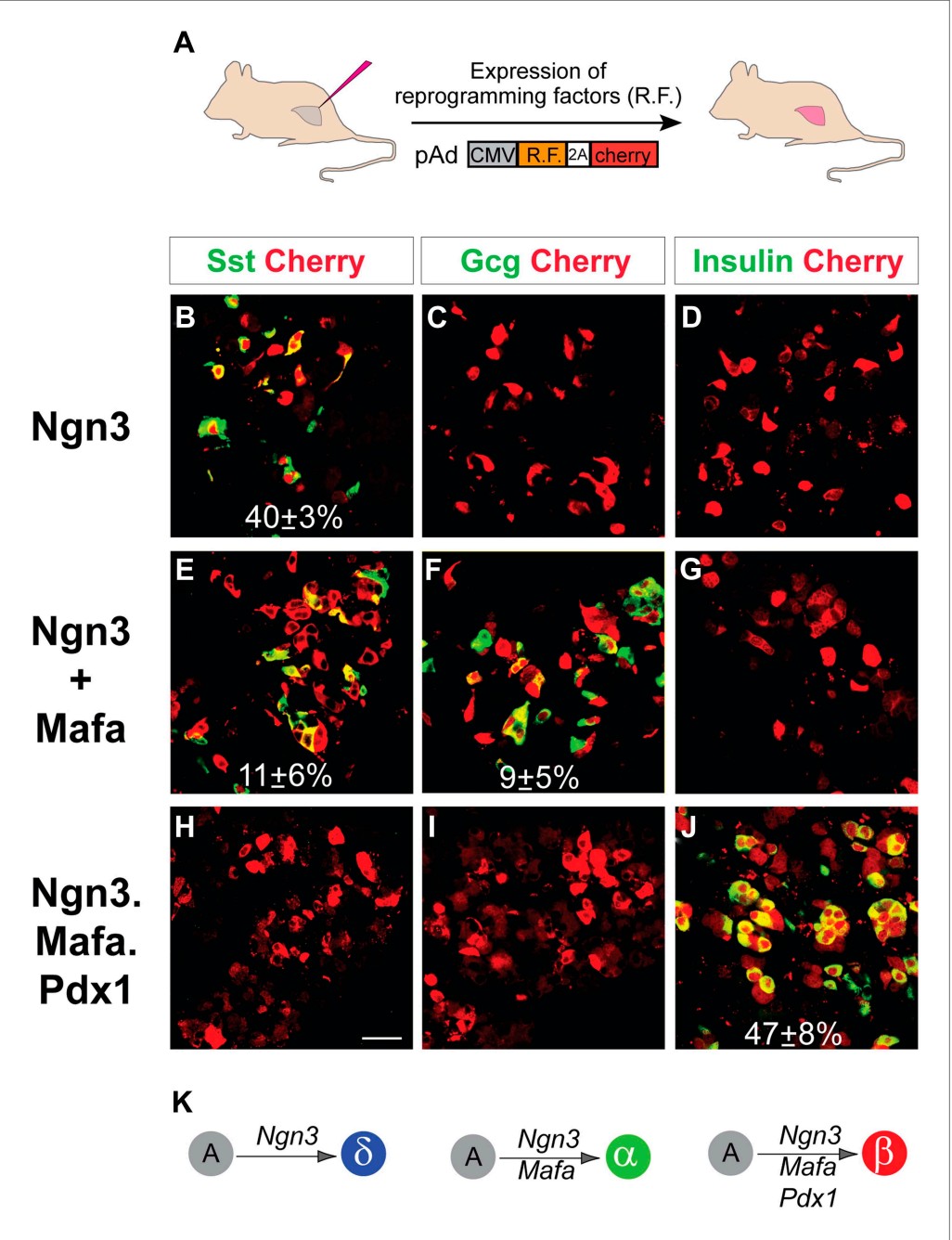

**Figure 1**. Induction of somatostatin[+], glucagon[+], and insulin[+] cells with defined factors in adult mouse pancreas in vivo. (**A**) Schematic diagram of experimental strategy. Adenoviruses co-expressing reprogramming factor (R.F.) and mCherry (cherry) were used to directly induce conversion of acinar cells in adult pancreas. 2A peptide that mediates polycistronic expression. Phenotypes were analyzed 10 days after induction. (**B–D**) Expression of Ngn3 alone induced 40 ± 3% of the infected mCherry[+] cells to become somatostatin[+] (Sst). (**E–G**) Co-infection of two separate viruses carrying Ngn3 and Mafa resulted in the formation of both glucagon[+] (Gcg) and somatostatin[+] cells in 11 ± 6% and 9 ± 5% of infected cells, respectively. (**H–I**) Co-expression of Ngn3, Mafa, Pdx1, and mCherry from a single polycistronic construct led to exclusive formation of insulin[+] cells in 47 ± 8% of the mCherry[+] cells. (**K**) Summary of pancreatic acinar cell conversion to endocrine subtypes with different combinations of factors. A, acinar cells. Quantifications are shown in mean ± s.d. At least 1000 cherry[+] cells counted from three different animals. Scale bar: 50 μm.

*Figure 1. Continued on next page*

*Figure 1. Continued*

The following figure supplements are available for figure 1:

**Figure supplement 1**. Adenoviral constructs used in the experiments and polycistronic factor expression.

**Figure supplement 2**. Mafa alone, Pdx1 alone, and combinations of Pdx1.Mafa and Ngn3.Pdx1 do not induce endocrine cells in pancreas.

**Figure supplement 3**. Transgene expression mediated by adenoviral infection in adult pancreas is transient.

and induced δ-cells after staining with somatostatin (*Figure 2—figure supplement 4*) (*Pechhold et al., 2009*). We note that this protocol allows isolation of genomic DNA but not intact mRNA from the pancreatic tissues. We have not been successful at generating gene-profiling data from the induced δ-cells using the intracellular FACS method.

None of the CpG sites assayed in the *somatostatin* promoters was methylated in all samples tested (*Figure 2—figure supplement 5*), indicating that this promoter is not subject to regulation by DNA methylation. *Amylase2,* a gene exclusively expressed in pancreatic acinar cells, was largely unmethylated in acinar cells (*Figure 2H*). In contrast, the induced δ-cells showed strong methylation in this promoter similar to islet δ-cells (*Figure 2H* and *Figure 2—figure supplement 6*). The *Insulin2* promoter was fully methylated in acinar cells but partially demethylated in both endogenous and induced δ-cells (*Figure 2H*). For both amylase and Insulin promoters, the methylation differences of acinar/islet δ-cell and acinar/induced δ-cell are statistically highly significant (p<0.001) whereas there is no significant difference between islet δ-cells and induced δ-cells (p=0.23 and 0.30 for amylase and insulin promoter respectively). These studies suggest that substantial DNA methylation changes occurred during acinar to δ-cell conversion in the promoters we studied. It is notable that not all cell fate conversion events are associated with DNA methylation changes. For example, no significant DNA methylation was observed in the conversion of pre-B cells to macrophages (*Rodriguez-Ubreva et al., 2012*).

We evaluated the ability of induced δ-cells to secret hormones in an in vitro secretion assay. The acinar fraction that contains induced δ-cells was isolated 30 days after induction and stimulated with the secretagogue Arginine. The induced δ-cells responded to Arginine and released somatostatin, in a manner similar to isolated islets which contain endogenous δ-cells (*Figure 2I*). In contrast, control acinar cells did not respond to Arginine stimulation (*Figure 2I*), consistent with the fact that they do not express endocrine hormones. These data suggest that induced δ-cells possess cellular machineries necessary for hormone production, storage, and release.

The data described above collectively indicate that δ-like cells induced by Ngn3 expression in adult pancreas possess key features of endogenous δ-cells.

## Ngn3 and Mafa converts acinar to α-like cells

Co-infection of adult mouse pancreas with two different adenoviruses carrying Ngn3- and Mafa-induced formation of glucagon[+] cells (*Figure 3A*). This co-infection also induced somatostatin[+] cells, which are distinct from the glucagon[+] cells (*Figure 3A'*). We tested different ratios of Ngn3/Mafa viruses and observed that a 1:1 ratio yielded the most number of Gcg[+] cells (*Figures 1F* and 9 ± 5% at 1:1 ratio, and data not shown). Due to the random nature of co-infection, cells that received predominately Ngn3 infection likely become the Sst[+] cells. In contrast to co-infection by two separate viruses, polycistronic co-expression of Ngn3 and Mafa from a single construct yielded substantially reduced number of glucagon[+] cells (less than 1%, data not shown). We therefore used co-infection to induce glucagon[+] cells in all subsequent experiments.

The most important gene that controls endogenous α-cell development and identity is the transcription factor Arx. Genetic deletion of Arx during embryonic development results in the absence of α-cells, whereas ectopic expression of Arx in β-cells causes their phenotypic drift towards α-cells (*Collombat et al., 2003*; *Dhawan et al., 2011*). We observed strong Arx expression in all induced Gcg[+] cells, although some Gcg[-] cells also expressed Arx (*Figures 3B,B'*). In addition, all Gcg[+] cells expressed the endocrine genes Pax6 (*Figure 3C,C'*) and synaptophysin (*Figure 3D,D'*). Due to the intermingling of induced Gcg[+] cells and Sst[+] cells, we were unable to purify these cells for detailed transcriptome analysis. The induced α-cells are readily observable 2 month after induction

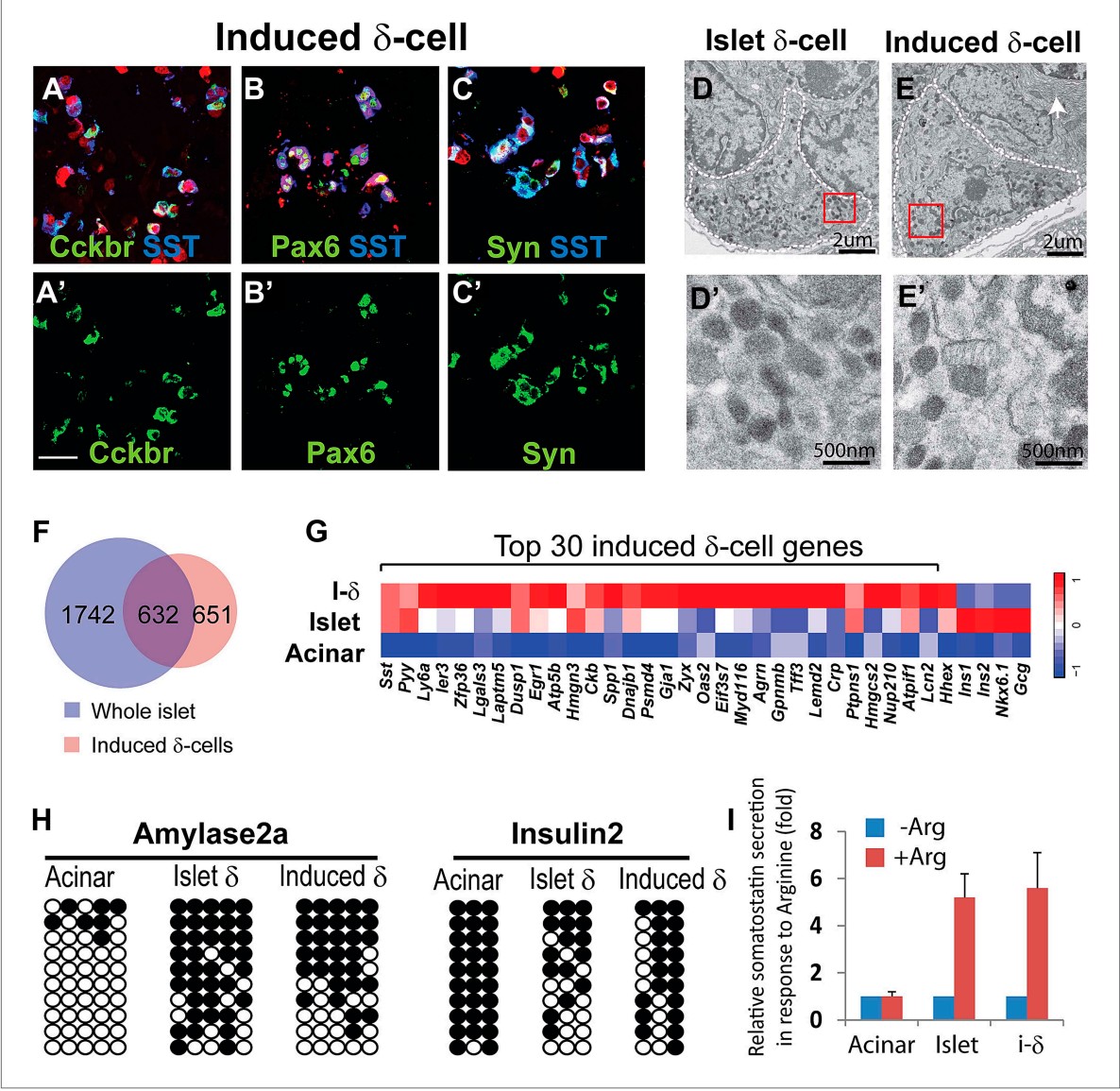

**Figure 2**. δ-like cell induction by Ngn3. (**A**–**C**) Induced δ-cells co-express somatostatin (SST) and cholecystokinin receptor B (Cckbr) (**A** and **A'**). They also co-express the endocrine markers Pax6 (**B** and **B'**) and Synaptophysin (Syn, **C** and **C'**). Scale bar: 50 μm. (**D** and **E**) Ultrastructure of endogenous and induced δ-cells in electron micrographs. **D'** and **E'** are magnified view of the boxed areas in **D** and **E**, showing the characteristic morphology of δ-cell granules. White arrow indicates a neighboring acinar cell with dense ER (endoplasmic reticulum) assembly. Induced δ-cells were found intermingled among acinar cells. In comparison, endogenous δ-cells reside exclusively in islets. (**F** and **G**) Transcriptional profiling identified 1283 genes enriched in induced δ-cells 30 days after induction. 632 of the induced genes are present in a whole-islet gene signature (**F**). Many of the top 30 induced δ-cell genes show medium to low expression in whole islet samples, which contain mostly β-cells. β- and α-specific genes, including *Ins1 (insulin1)*, *Ins2 (insulin2)*, *NKX6.1*, and *Gcg (glucagon)*, are absent from the induced δ-cell samples. (**H**) DNA methylation analysis of the proximal promoters of *Amylase 2a* and *Insulin2* genes in acinar cells, islet δ-cells, and induced δ-cells (20 days after induction). Methylation status of the induced and endogenous δ-cells is similar, indicating appropriate methylation changes during acinar to δ-cell conversion. (**I**) Induced δ-cells released somatostatin in response to the secretagogue Arginine (20 mM) in an in vitro assay. Acinar cells and islets were used as controls. Data were normalized as fold increase over baseline (no Arginine). Quantifications are shown in mean ± SD, n = 3 animals.

The following figure supplements are available for figure 2:

**Figure supplement 1**. Induced δ-cells persist in adult pancreas.

**Figure supplement 2**. Ultrastructure comparison of induced and endogenous endocrine subtypes.

*Figure 2. Continued on next page*

*Figure 2. Continued*

**Figure supplement 3**. Genomic maps of CpG sites in the promoter region of mouse *insulin2 (ins2)* and *amylase2a2 (Amy2a2)* genes.

**Figure supplement 4**. Purification of endogenous δ- and α-cells, and induced δ-cells by intracellular FACS for DNA methylation studies.

**Figure supplement 5**. *Somatostatin* promoter DNA methylation analysis.

**Figure supplement 6**. COBRA analysis of *Amylase* promoter.

(*Figure 3—figure supplement 1*). Electron microscopy analysis of the induced α-cells indicates that they share a similar ultrastructure as the endogenous α-cells (*Figure 3E,F*). The secretory granules have a narrow halo between the core and membrane (*Figure 3E',F'*) that is typical of α-granules and distinct from δ- and β-granules (*Figure 2—figure supplement 2*). The induced α-like cells were intermingled among acinar cells (*Figure 3F*, arrow points to zymogene granules of a neighboring acinar cells), whereas endogenous α-cells were found exclusively within islets (*Figure 3E*). In an in vitro hormone secretion assay, the acinar fraction that contains induced α-cells isolated from the adult pancreas 30 days after induction responded to the secretagogue Arginine and released glucagon (*Figure 3H*), indicating their ability to produce and secrete hormone.

We used intracellular FACS to isolate endogenous and induced α-cells (*Figure 2—figure supplement 4* and *Figure 3—figure supplement 2*). DNA methylation analysis at the *Glucagon, Amylase2a,* and *Insulin2* promoters showed a general similarity of induced α-cells with endogenous α-cells (*Figure 3G*, *Figure 3—figure supplement 3*). Statistical analysis showed no significant difference between endogenous and induced cells at glucagon and insulin promoters (p=0.26, 0.50, respectively). However, a difference was detected at the *amylase* promoter (p=0.02). Amylase protein expression was absent in the majority of Gcg+ cells (over 95%). However, a small fraction (<5% of all Gcg+ cells) were Amylase+ (*Figure 3—figure supplement 4*), indicating incomplete reprogramming in a small subset of glucagon+ cells.

Taken together, these results indicate that induced α-cells possess key features of endogenous α-cells.

## Induced δ- and α-cells are converted from adult acinar cells in the absence of cell proliferation

We previously reported that the acinar cell is the cell-of-origin for most of the induced β-cells in M3 factor-mediated reprogramming of adult pancreas (*Zhou et al., 2008*). This is in part due to preferential infection of adenovirus for acinar cells but not the other pancreatic cell types (*Zhou et al., 2008*). To test whether the induced δ-and α-cells also derive from acinar cells, we used a genetic lineage tracing strategy similar to the previous study. A Ptf1aCreER mouse line, which drives CreER expression exclusively in acinar cells of the adult pancreas, was crossed with a Rosa-floxed-Stop-YFP (RosaYFP) reporter line to create bigenic Ptf1aCreER::RosaYFP animals. Tamoxifen induction resulted in the labeling of 20–30% of mature acinar cells (*Figure 4A',B'*), consistent with published report of this line (*Pan et al., 2013*). After adenoviral delivery of Ngn3 or Ngn3+Mafa, which targets acinar cells, we confirmed that acinar cells can give rise to both δ-like and α-like cells (*Figure 4A,B*). The induced endocrine cells are also smaller than acinar cells (*Figure 4A,B*, arrows), consistent with previous report (*Zhou et al., 2008*).

Continuous BrdU labeling showed that the conversion of acinar cells to induced δ- and α-cells occurred largely in the absence of cell proliferation (*Figure 4C,D*). Only about 1% of the induced δ- and α-cells incorporated BrdU in a 10-day reprogramming period (*Figure 4E*).

These studies confirmed that induced δ- and α-cells derive from adult acinar cells and that the acinar conversion occurred largely in the absence of cell proliferation.

## Ngn3 promotes an endocrine state in acinar cells by suppressing acinar fate regulators and activating pan-endocrine genes

In principle, the conversion of one cell fate to another involves two major components: suppression of the original cell fate and activation of a new one within the same cell. We examined the ability of the

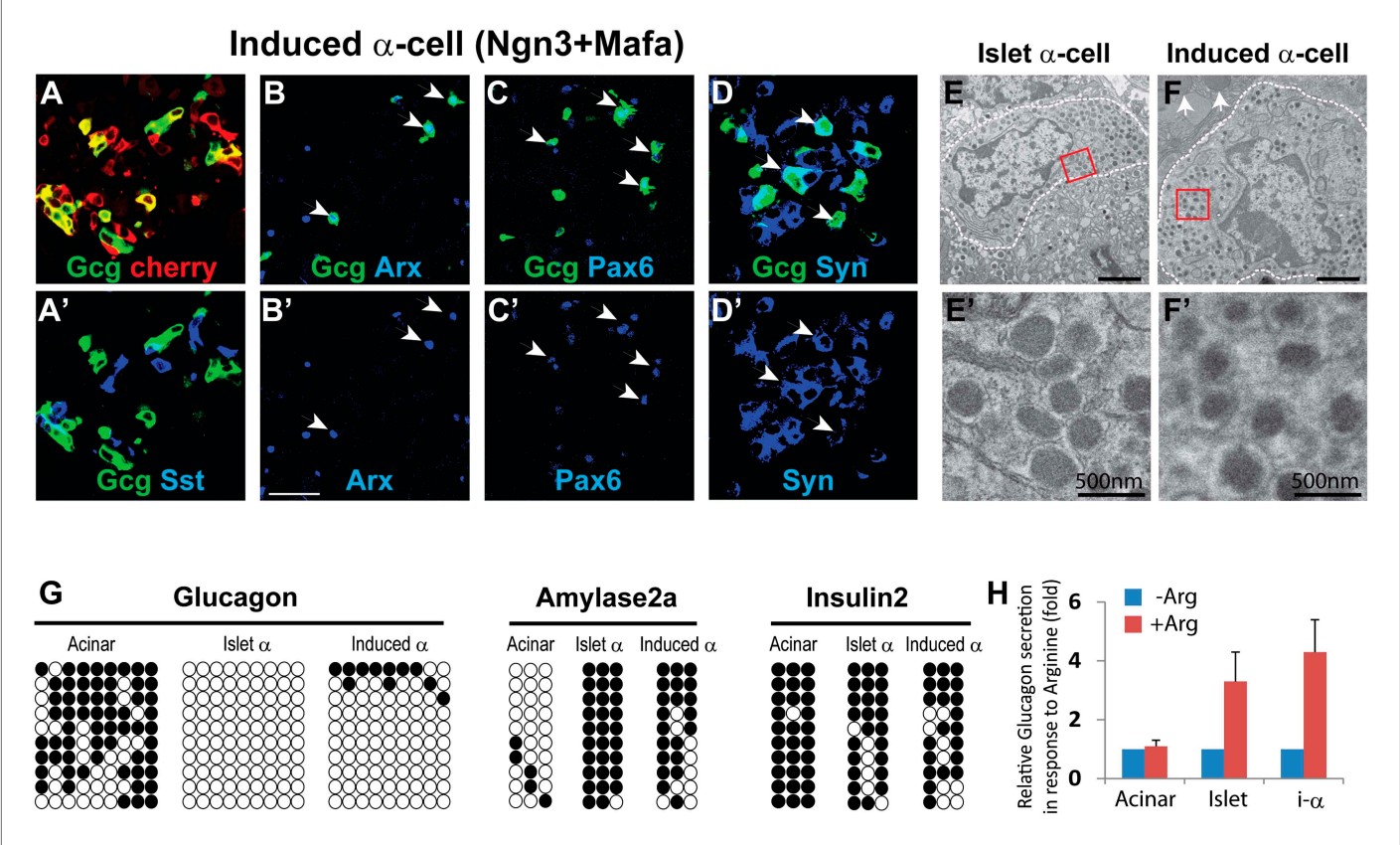

**Figure 3**. α-like cell induction by Ngn3 and Mafa. (**A**) Co-infection of two separate viruses carrying Ngn3 and Mafa led to the induction of Glucagon (Gcg⁺) cells. Somatostatin (Sst⁺) cells were also induced as a separate population (**A'**). (**B–D**) Induced Gcg⁺ cells express α-cell fate regulator Arx (**B** and **B'**) and endocrine factors Pax6 (**C** and **C'**) and synaptophysin (Syn, **D** and **D'**). Arrows indicate double positive cells. Scale bar: 50 μm. Syn is expressed in both Gcg⁺ and Sst⁺ cells. (**E** and **F**) Electron micrographs of endogenous and induced α-cells. **E'** and **F'** are magnified view of the boxed areas in **E** and **F**, showing the characteristic morphology of α-cell granules. Arrows indicate zymogen granules of a neighboring acinar cell. Endogenous α-cells reside within islets, whereas induced α-cells reside among acinar cells. (**G**) DNA methylation analysis of the proximal promoters of *Glucagon*, *Amylase 2a*, and *Insulin2* genes in acinar cells, islet α-cells, and induced α-cells (20 days after induction). Methylation status of the induced and endogenous α-cells is similar, indicating appropriate methylation changes during acinar to α-cell conversion. (**H**) Induced α-cells responded to stimulation by the secretagogue Arginine (20 mM) and released glucagon. Acinar and islets were used as controls. Data were normalized as fold increase over baseline (no Arginine). Quantifications are shown in mean ± s.d., n = 3 animals.

The following figure supplements are available for figure 3:

**Figure supplement 1**. Induced α-cells persist in adult pancreas.

**Figure supplement 2**. Purification of induced α-cells by intracellular FACS for DNA methylation studies.

**Figure supplement 3**. Genomic map of CpG sites in the promoter region of mouse *glucagon* gene.

**Figure supplement 4**. A small number of Gcg⁺ cells are partially reprogrammed.

three reprogramming factors to suppress acinar fate-regulators and activate pan-endocrine genes, two key events necessary for establishing an endocrine fate in acinar cells.

Ptf1a, Nr5a2, and Mist1 are key acinar cell-fate regulators. They are expressed in adult acinar; their genetic deletion results in abnormalities of acinar development and function (*Pin et al., 2001*; *Kawaguchi et al., 2002*; *Lin et al., 2004*; *Beres et al., 2006*; *Holmstrom et al., 2011*). We observed that Ngn3 and Mafa, but not Pdx1, strongly suppressed Ptf1a, Mist1, and Nr5a2 expression 4 days after gene delivery in pancreas (*Figure 5A,B,E,F,I,J,M*).

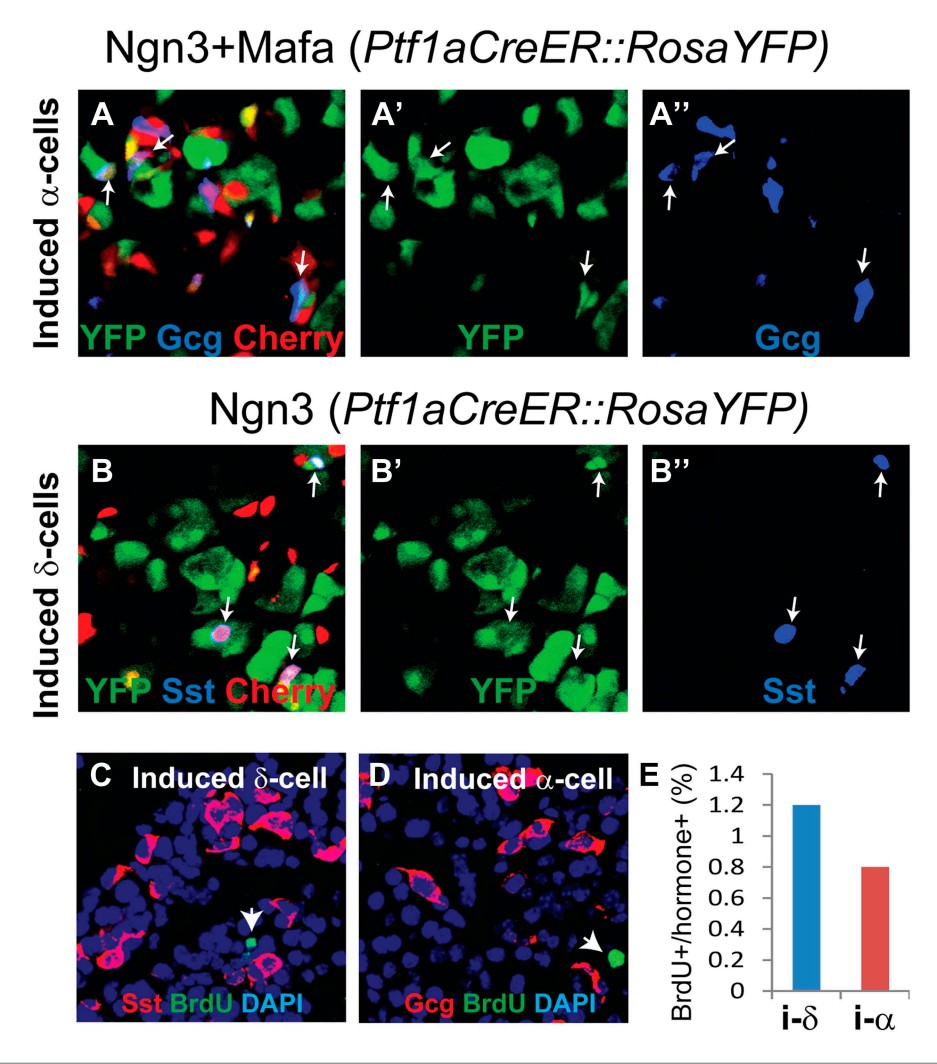

**Figure 4**. Induced δ- and α-like cells are converted from acinar cells in the absence of cell proliferation. (**A–B**) Genetic lineage tracing of induced δ- and α-like cells. Tamoxifen induction of bigenic *Ptf1aCreER::RosaYFP* animals led to specific and indelible labeling of approximately 20% of adult pancreatic acinar cells. Delivery of Ngn3+Mafa or Ngn3 by adenovirus in the pancreas resulted in formation of Gcg+YFP+Cherry+ (**A–A''**, arrows) and Sst+YFP+Cherry+ (**B–B''**, arrows) cells, indicating that the induced cells derive from adult acinar cells. Note that both endogenous and induced endocrine cells are smaller than acinar cells. (**C–E**) Continued BrdU labeling during the first 10 days of δ- and α-induction showed that few induced cells incorporated BrdU, indicating a lack of proliferation during this period. Arrows indicate BrdU+ cells. A total of 1000 Sst+ or Gcg+ cells were quantified from three animals. i-δ: induced δ-cells. i-α: induced α-cells.

*Pax6* and *Islet1* are endocrine genes expressed in all islet endocrine cells. Genetic studies have established their important role in the development of all islet endocrine subtypes (*Ahlgren et al., 1997*; *Sander et al., 1997*; *Ashery-Padan et al., 2004*). Ngn3, but not Mafa or Pdx1, activated the expression of these pan-endocrine genes at day 4 after pancreas infection (*Figure 5C,D,G,H,K,L,M*).

The data discussed above together indicate that among the three reprogramming factors, Ngn3 alone possesses the ability to initiate two key events in endocrine reprogramming, namely, acinar suppression and pan-endocrine activation, thereby establishing a generic endocrine state in acinar cells.

## Acinar factors are molecular barriers of endocrine reprogramming

Given the important role played by key acinar factors in maintaining acinar cell fate, we hypothesized that these factors could act as molecular barriers in endocrine conversion. We tested this hypothesis

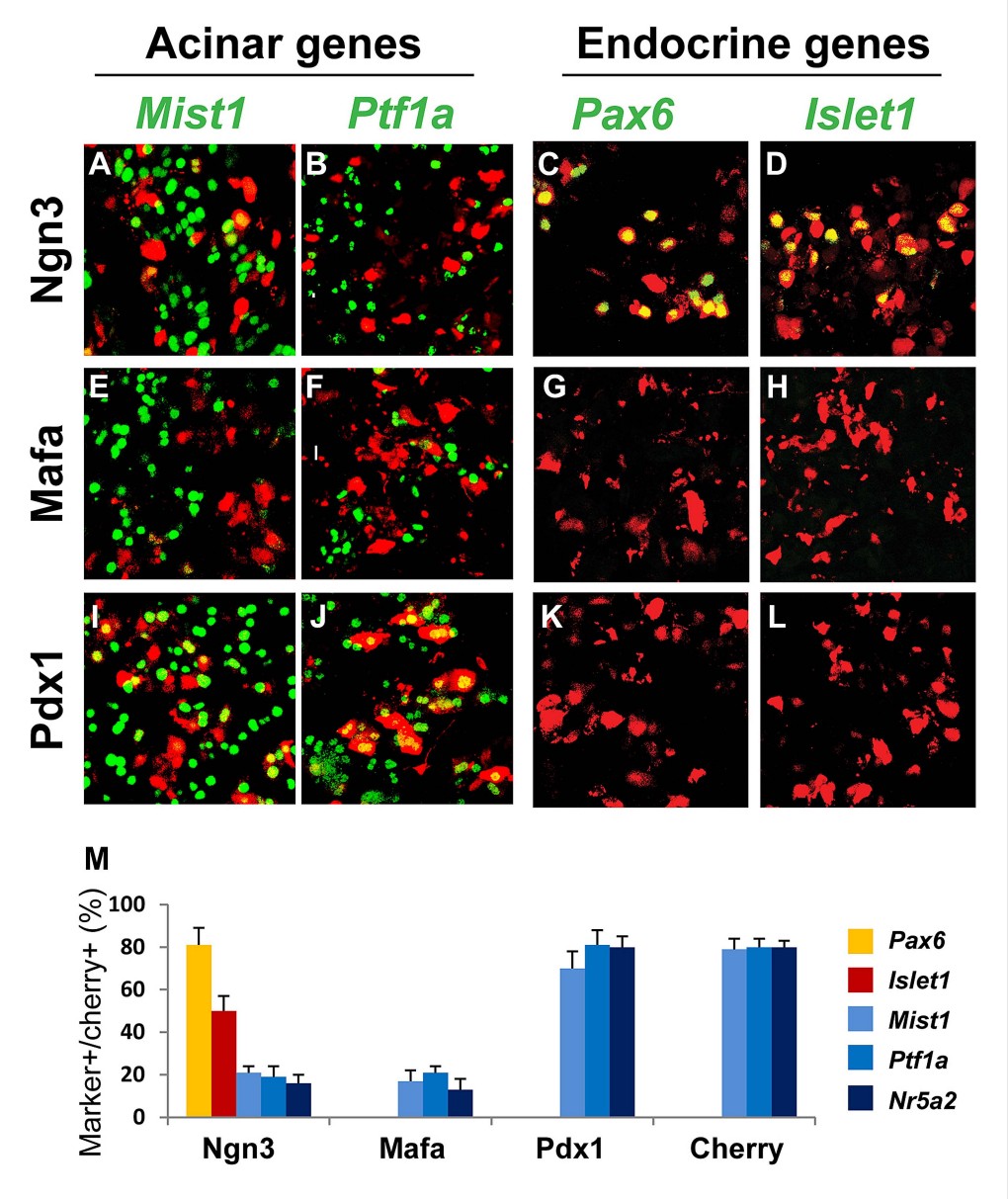

**Figure 5**. Ngn3 can simultaneously suppress acinar fate-regulators and activate pan-endocrine genes to establish an endocrine state. Immunohistochemistry showed that 4 days after expression of the three reprogramming factors individually in the pancreas, Ngn3 and Mafa, but not Pdx1, strongly suppressed the expression of the acinar fate-regulators Mist1, Ptf1a, and Nr5a2 (**A**, **B**, **E**, **F**, **I**, **J**, **M**). Ngn3 also activated expression of the pan-endocrine genes Pax6 and Islet1 (**C**, **D**, **M**), whereas Mafa and Pdx1 did not (**G**, **H**, **K**, **L**, **M**). Ngn3 alone can therefore establish an endocrine state in acinar cells by simultaneous suppression of acinar factors and activation of pan-endocrine genes. Infection with Cherry was used as control (**M**). Quantifications are shown as mean ± s.d. At least 1000 cherry[+] cells counted from three different animals.

by co-infecting pancreas with two viruses carrying *Nr5a2* and *Ngn3*. 10 days after the infection, analyses revealed that persistent expression of Nr5a2 strongly blocked induction of Pax6 and Sst (***Figure 6B,E,G***), compared with Ngn3 alone controls (***Figure 6A,D,G***). Another acinar factor Ptf1a similarly suppressed Pax6 and Sst induction (***Figure 6C,F,G***). These results demonstrate that key acinar factors are potent molecular barriers; their down-regulation is a prerequisite for endocrine reprogramming.

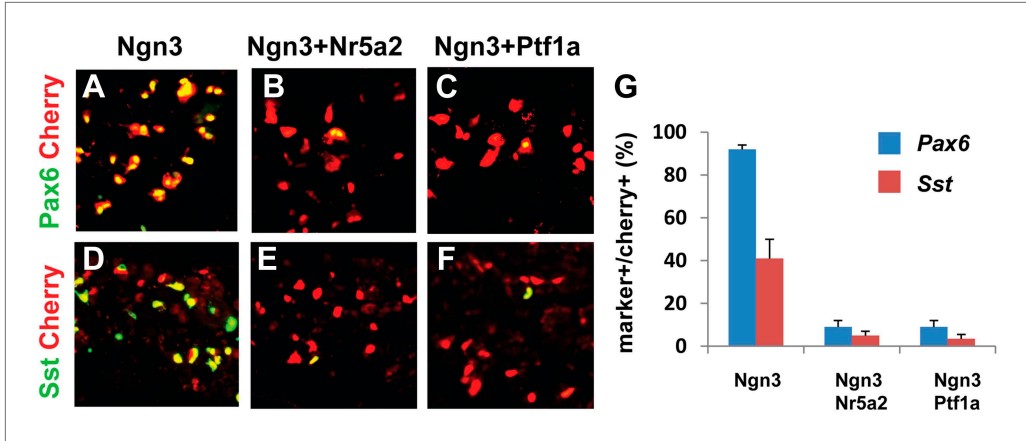

**Figure 6**. Acinar factors are molecular barriers of endocrine reprogramming. Compared with the robust induction of Pax6 and Sst by Ngn3 alone (**A**, **D**, **G**), co-expression of Nr5a2 and Ngn3 (by co-infection of two separate viruses) strongly inhibited the activation of both endocrine genes (**B**, **E**, **G**). A similar suppression was observed when Ngn3 was co-expressed with Ptf1a (**C**, **F**, **G**). Samples were analyzed 10 days after infection. Quantifications are shown as mean ± s.d. At least 1000 cherry$^+$ cells counted from three different animals.

## Establishment of endocrine state precedes activation of endocrine subtype-specific genes in acinar conversion

To further understand the temporal sequence of acinar to endocrine subtype conversion, we examined three key events: acinar suppression, pan-endocrine activation, and subtype-specific gene activation. In Ngn3-induced acinar to δ-cell conversion, we observed strong suppression of the acinar factor Mist1 at day 2 after Ngn3 delivery (**Figure 7A**). The pan-endocrine gene Pax6 was also robustly activated at day 2 (**Figure 7B**). In contrast, expression of δ-specific genes somatostatin and CCKbr was not detected until day 4 (**Figure 7G,H**) and became strongly expressed at day 10 (**Figure 7K,L**). A similar temporal sequence of acinar suppression (Mist1) and pan-endocrine activation (Pax6) followed by β-specific gene activation (Nkx6.1 and insulin) was also observed in acinar to β-cell conversion (**Figure 8**). These data collectively suggest that a generic endocrine state was established in acinar cells at the onset of reprogramming, temporally preceding endocrine subtype specification.

### Pdx1 and Mafa can suppress δ-subtype specification

The studies discussed above indicate that Ngn3 has two main functions in acinar to endocrine conversion: establishment of an endocrine state, and δ-subtype specification in the absence of other factors. Ngn3 is also part of the reprogramming factor combination in the induction of α-cells (Ngn3+Mafa) and β-cells (Ngn3+Mafa + Pdx1), which raises the question of how a singular α- or β-cell fate is established. We tested the possibility that δ-specification may be suppressed by Pdx1 and/or Mafa. Polycistronic co-expression of Mafa with Ngn3 reduced the efficiency of somatostatin induction from 40 ± 3% in control Ngn3 animals (**Figure 9A**) to 24 ± 7% in Ngn3.Mafa animals (**Figure 9B,D**). Further increasing the amount of Mafa in the mixture led to a stronger suppression of δ-cell induction (data not shown). Polycistronic co-expression of Pdx1 with Ngn3 led to near complete suppression of somatostatin$^+$ cells (2 ± 1%, **Figure 9C,D**). Immunostaining with pan-endocrine factor synaptophysin revealed that Cherry$^+$somatostatin$^-$ cells were synaptophysin$^+$ (**Figure 9C**, inset), suggesting that Ngn3 converted acinar cells to an endocrine state, but that δ-subtype specification was blocked. These data indicate that Pdx1 and Mafa can suppress δ-subtype specification, which is likely part of the mechanism to ensure formation of distinct α- and β-subtypes upon coexpression of multiple reprogramming factors.

## Discussion

Our studies indicate that pancreatic acinar cells can be directly converted to endocrine δ- and α-like cells by in vivo expression of Ngn3 or Ngn3+Mafa respectively. Together with our previous report of β-cell reprogramming with Ngn3+Mafa+Pdx1, these studies provide a set of reprogramming models

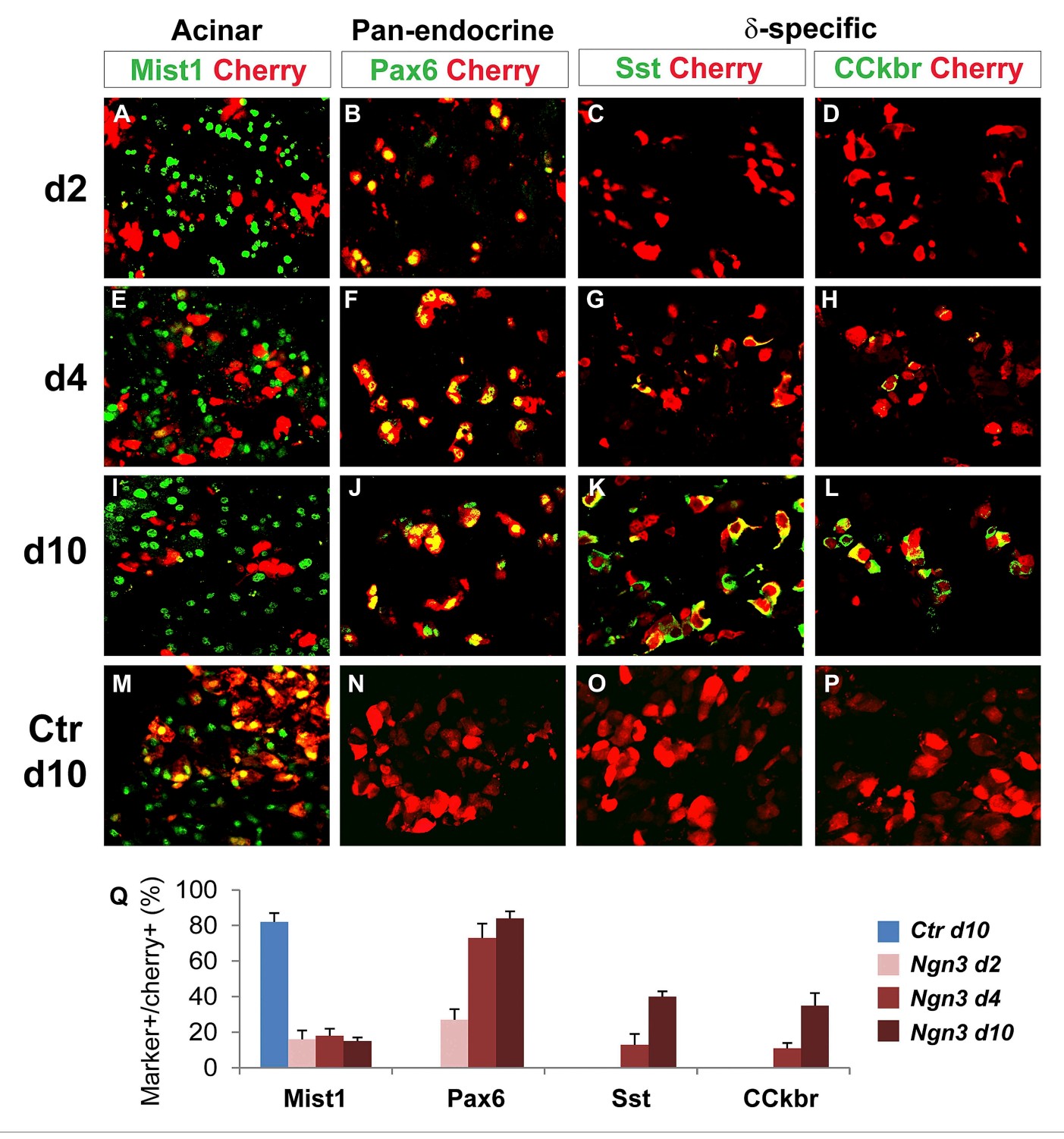

**Figure 7**. Acinar suppression and pan-endocrine activation precedes subtype-specific gene activation in acinar to δ-cell conversion. In acinar to δ-cell conversion induced by Ngn3, strong suppression of the acinar factor Mist1 was observed in the Cherry⁺-infected cells at day 2 (**A**). The pan-endocrine factor Pax6 was also induced at day 2 (**B**). The Mist1⁻Pax6⁺ state was maintained in the majority of Cherry⁺ cells at later time points (**E**, **F**, **I**, **J**). In contrast, δ-subtype specific factors Sst and CCkbr were not induced until day 4 (**G**, **H**) and became robustly expressed at day 10 (**K**, **L**). In control samples expressing cherry alone, the majority of cherry⁺ acinar cells had Mist1 expression (**M**), and none had induced endocrine gene expression (**N**, **O**, **P**). Quantifications are shown as mean ± s.d (**Q**). At least 1000 cherry⁺ cells counted from three different animals.

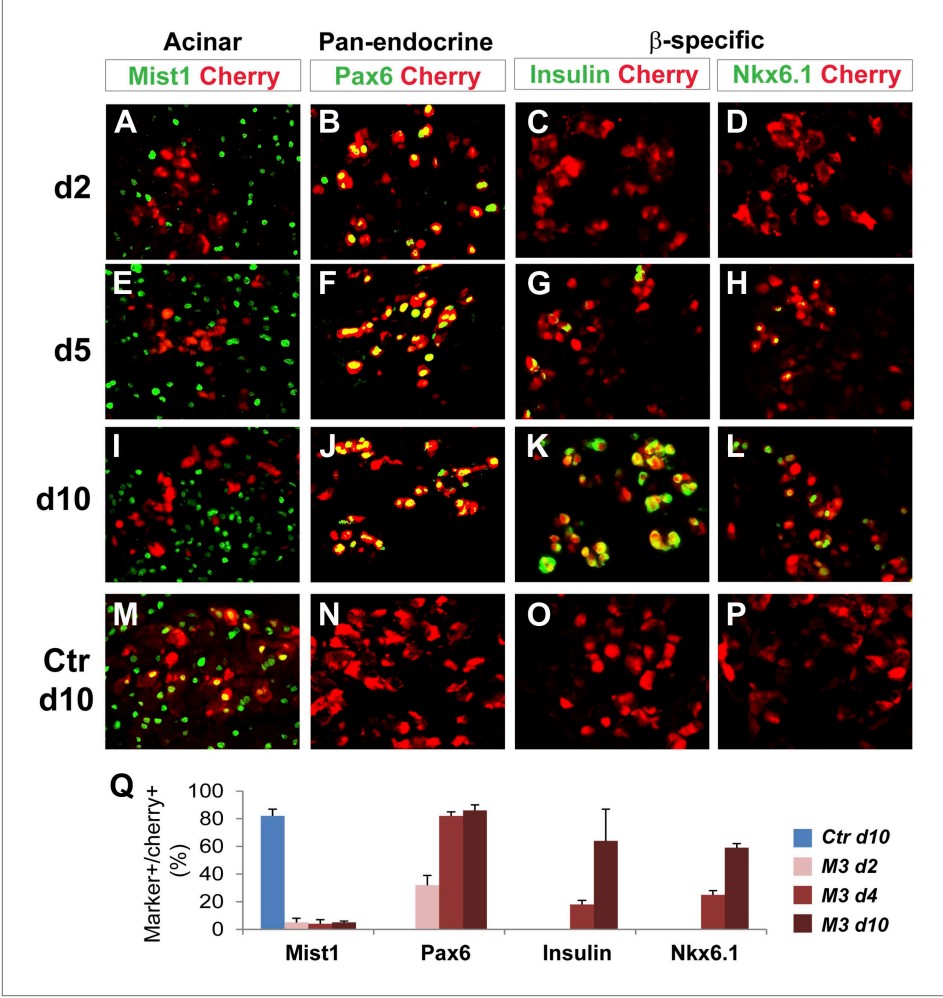

**Figure 8**. Acinar suppression and pan-endocrine activation precedes subtype-specific gene activation in acinar to β-cell conversion. In acinar to β-cell conversion induced by M3 (Ngn3+Mafa+Pdx1), near complete suppression of the acinar factor Mist1 was observed in the Cherry⁺-infected cells at day 2 (**A**). The pan-endocrine factor Pax6 was also robustly induced at day 2 (**B**). The Mist1⁻Pax6⁺ state was maintained in the majority of Cherry⁺ cells at later time points (**E**, **F**, **I**, **J**). In contrast, β-subtype specific factors insulin and Nkx6.1 were not induced until day 5 (**G**, **H**) and became more robustly expressed at day 10 (**K**, **L**). In control samples expressing cherry alone, the majority of cherry⁺ acinar cells had Mist1 expression (**M**), and none had induced endocrine gene expression (**N**, **O**, **P**). Quantifications are shown as mean ± s.d (**Q**). At least 1000 cherry⁺ cells counted from three different animals.

where combinatorial actions of three factors lead to conversion of acinar cells to the three major islet endocrine subtypes in vivo (*Figure 10A*).

Our studies indicate that acinar to endocrine reprogramming includes two main processes: establishment of a generic endocrine state and endocrine subtype specification (*Figure 10B*). Ngn3 plays a central role in promoting the endocrine state at the onset of reprogramming by suppressing acinar fate regulators and activating pan-endocrine genes (*Figure 10B*, upper panel). This early step is critical as continued expression of key acinar factors will block endocrine conversion.

Compared with Ngn3, Mafa suppresses acinar regulators but does not activate pan-endocrine genes, whereas Pdx1 lacks the ability to initiate either of these two events. It is likely that the main function of Mafa and Pdx1 is to collaborate with Ngn3 to activate subtype-specific genes such as Arx in α-induction (this study) and Nkx6.1 in β-induction (*Zhou et al., 2008*). Another important function of Mafa and Pdx1 is to suppress δ-specification, thus ensuring formation of singular α- and β-subtypes, and preventing hybrid cells (*Figure 10B*, lower panel). Our data together suggest that each of the

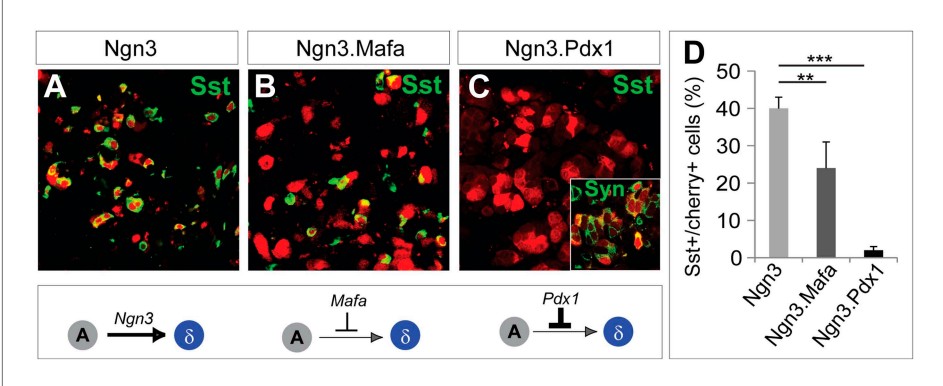

**Figure 9**. Pdx1 and Mafa can suppress δ-subtype specification. Compared with robust induction of Sst$^+$ cells by Ngn3 alone (**A**), polycistronic co-expression of Mafa and Ngn3 led to strong reduction of Sst$^+$ cells (**B**). Polycistronic co-expression of Pdx1 and Ngn3 nearly completely suppressed Sst$^+$ cell induction (**C**). The Sst$^-$cherry$^+$ cells expressed synaptophysin (Syn) (**C**, inset), suggesting that these cells acquired an endocrine identity but δ-specification was blocked. Quantifications are shown as mean ± s.d (**D**). At least 1000 cherry$^+$ cells counted from three different animals. **p<0.01, ***p<0.001. Mann–Whitney test.

three factors possesses both activator and suppressor functions (*Figure 10C*). It will be important to fully elucidate the molecular details of these functions in future studies.

During pancreatic development, numerous studies have highlighted the importance of Ngn3 in controlling endocrine fate specification (*Jensen, 2004*). A recent study showed that ectopic expression of Ngn3 in embryonic pancreas suppressed exocrine fate specification (*Qu et al., 2013*). Thus, Ngn3 can suppress exocrine fate and promote endocrine fate during both embryogenesis in progenitor cells and during adulthood in differentiated acinar cells. The function of Pdx1 has also been extensively studied and demonstrated to be critical in early pancreatic fate determination, as well as a later role in beta cell biology (*Murtaugh, 2007*). It is not clear, however, exactly what role Pdx1 plays in δ- and α-cell specification in embryogenesis. Mafa is expressed in pancreatic β-cells during development and plays a role in controlling β-cell gene transcription (*Hang and Stein, 2011*). The ability of Mafa to induce α-cell formation in collaboration with Ngn3 from acinar is nevertheless not entirely surprising. Its close homologue Mafb is expressed during pancreas development in a subset of endocrine progenitor cells (*Nishimura et al., 2006*). Mafb deletion leads to decreased α- and β-cell numbers, suggesting a role for Maf factor in α-cell formation (*Artner et al., 2007*; *Nishimura et al., 2008*). Taken together, there are clear similarities in the function of Ngn3, Mafa, and Pdx1 in normal endocrine development and endocrine reprogramming from acinar cells. Nevertheless, the epigenetic landscape of pancreatic progenitors and adult acinar cells in which these factors operate is presumably very different. Further, the 'generic endocrine state' established by Ngn3 in acinar cells is not equivalent to an endocrine progenitor in pancreas development. For example, expression of pancreatic progenitor genes such as *Sox9* and *Hnf6* was not detected during the reprogramming process (data not shown).

One surprising finding from our study is that Ngn3 alone promotes δ-cell formation from acinar cells. Among the pancreatic endocrine subtypes, relatively little is known about the δ-cells. No genetic factors have been discovered that specify δ-cell fate in pancreas development. The molecular details of how Ngn3 functions to specify δ-subtype remains to be determined.

Conversion of acinar cells to α-cells requires both Ngn3 and Mafa. The optimal method to induce Gcg$^+$ α-like cells is by co-expression of Ngn3 and Mafa from two separate viruses at a 1:1 ratio. Varying the ratio of the two viruses or polycistronic co-expression of both factors led to reduced α-cell formation. These results are reminiscent of induced pluripotent reprogramming from fibroblasts where polycistronic co-expression of OSKM factors led to substantially reduced efficiency of iPS formation compared with random infection by OSKM factors (*Carey et al., 2009*; *Chang et al., 2009*). This difference has been suggested to result from a need for appropriate reprogramming factor stoichiometry (*Carey et al., 2011*). We speculate that the induction of α-like cells may similarly require optimized Ngn3/Mafa stoichiometry along with other conditions such as specific expression level and dynamics of the factors, which can be met at the more 'flexible' co-injection system with each cell expressing

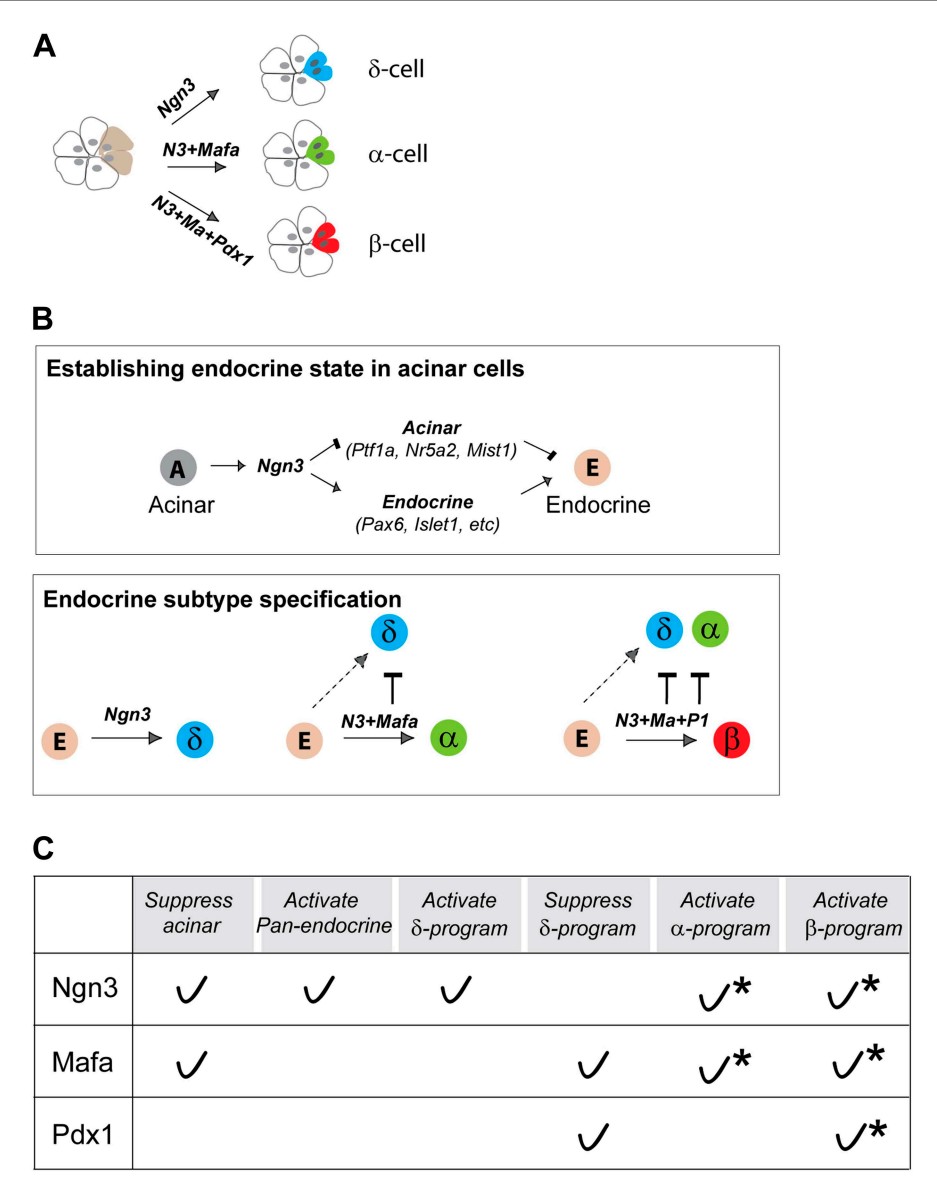

**Figure 10**. Direct in vivo conversion of pancreatic acinar cells to three islet endocrine subtypes by combinatorial actions of three factors. (**A**) Summary of acinar to islet endocrine conversion with defined factors. (**B**) Our studies suggest that there are two main processes in pancreatic acinar to endocrine reprogramming. Ngn3 plays a critical role in establishing a generic endocrine state in acinar cells by suppressing acinar fate regulators (Ptf1a, Nr5a2, Mist1) and activating pan-endocrine factors (Pax6, Islet1, etc) (upper panel). Down-regulation of acinar regulators is critical as they can block reprogramming. In endocrine subtype-specification (lower panel), Ngn3 promotes δ-fate in the absence of other factors. Mafa and Pdx1 act in concert with Ngn3 to promote α- and β-specification. Both Mafa and Pdx1 can suppress δ-subtype specification, whereas α-specification is also suppressed in β-induction. Combinatorial actions of the three reprogramming factors therefore led to formation of distinct endocrine subtypes. (**C**) A summary table of reprogramming factor functions. Asterisks: combinatorial actions of multiple factors are required to specify α- and β-cells from acinar cells.

variable levels of Ngn3/Mafa, compared with the more 'fixed' polycistronic expression system. Future experiments will be required to elucidate mechanisms of α-cell induction.

Our studies suggest that a suitable strategy to produce specific subtypes of cells by lineage conversion is to combine factors that confer broad competence with factors that confer subtype specificity. An elegant example was demonstrated in *C. elegans*, where removal of a chromatin factor and

employment of neuron selector genes allows conversion of germ cells to different neuronal subtypes (*Tursun et al., 2011*). The mammalian system is far more complex, but similar principles may well apply. It is hoped that insight from lineage reprogramming studies will lead to informed design and improved technology, thereby helping to unlock the tremendous therapeutic potential of this approach.

## Materials and methods

### Construction and purification of adenovirus

Genes of interest were first cloned into a shuttle vector containing a 2A-Cherry, then into the pAd/CMV/V5-DEST adenoviral vector (Invitrogen, Grand Island, NY). High titer virus (2–10 × $10^{10}$ pfu/ml) was obtained by purification with the Vivapure Adenopack (Sartorius, Bohemia, NY). Viral tittering was performed with direct immuno-staining of inserted genes (*Pdx1*, *Mafa*, *Ngn3*) 2 days after infection in HEK293A cells. Viral preparations that did not reach at least 2 × $10^{10}$ pfu/ml in one round of purification had poor induction efficiency in vivo, and were not used.

### Animals, surgery

*Rag1*$^{-/-}$ animals were obtained from Jackson Labs (Bar Harbor, ME). Adult animals (2–3 month) were injected with 100 µl of purified adenovirus (typically 1–2 × $10^9$ pfu, dilution with saline of high titer stocks) directly into the splenic lobe of the dorsal pancreas with a 3/10 cc Insulin Syringe (Becton Dickinson, East Rutherford, NJ). All experiments were performed under approved institutional regulations.

### Immunohistochemistry

Adult mouse pancreata were processed as previously described (*Zhou et al., 2008*). The following primary antibodies were used: goat anti-Ngn3 (Santa Cruz), guinea pig anti-Insulin (Dako, Carpinteria, CA), guinea pig anti-Glucagon (Linco, Charles, MO), rabbit anti-somatostatin (Dako), goat anti-Somatostatin (Santa Cruz), goat anti-Pdx1 (Santa Cruz, Dallas, TX), rabbit anti-mafA (Bethyl, Montgomery, TX), goat anti-Glut2 (Santa Cruz), rabbit anti-synaptophysin (Abcam, Cambridge, MA), rabbit anti-Ptf1a (BCBC), mouse anti-Mist1 (Santa Cruz), rabbit anti-Nkx6.1 (BCBC, Nashville, TN), rabbit anti-Sox9 (Santa Cruz), mouse anti-Pax6 (DSHB, Iowa City, IA), and mouse anti-islet (DSHB). Secondary antibodies were obtained from the Jackson Immunoresearch laboratories (West Grove, PA) and Life Technologies. Pictures were taken with a Zeiss LSM 510 META confocal microscope.

### Islet and acinar preparations, FACS of cherry⁺ cells, gene profiling

Standard procedures were used to separate adult mouse pancreas into islet fraction and exocrine fraction after intra-ductal perfusion and digestion with liberase (Roche, Indianapolis, IN). After Dithizone staining, the acinar fraction is manually picked to eliminate all visible islets. Islets and acinar clusters are further dissociated into single cells by EGTA treatment. Cherry⁺ cells were subsequently isolated by fluorescent activated sorting (FACS) with FACSaria (BD Bioscience, San Jose, CA). RNA was extracted (Qiagene RNeasy kit, Germantown, MD), cRNA synthesized (Ambion Amplification kit, Grand Island, NY), and genome-wide gene profiling performed with Illumina arrays.

### Intracellular FACS sorting of somatostatin⁺ and glucagon⁺ cells

Intracytoplasmic staining of pancreatic cells was performed as previously described (*Pechhold et al., 2009*) with minor modifications. Cells were fixed with 4% paraformaldehyde in PBS for 5 min on ice, diluted in wash buffer (WB) (1:10), centrifuged at 250×*g* for 5 min, and permeabilized with detergent wash buffer (WB(d)) for 30 min on ice. Primary antibodies and final concentration used for the intracytoplasmic staining are mouse monoclonal anti-Glucagon (K79bB10, Sigma St. Louis, MO; 1/1000) and Goat anti-Somatostatin (Santa Cruz; sc-7819; 1/500). All primary antibodies were pre-labeled with Alexa Fluor 594, Alexa Fluor 488, or Alexa Fluor 647, using Zenon antibody labeling kits according to the manufacture's protocol. Intracellular FACS was carried out with FACSaria (BD Bioscience).

### DNA methylation analysis, bisulphite sequencing, COBRA

Genomic DNA was purified using RecoverAll Total Nucleic Acid Isolation Kit (Invitrogen) and treated with EpiTect Bisulfite Kit (QIAGEN) according to the manufacture's protocols. The bisulfited genomic DNA was amplified using a touch-down PCR protocol (*Amylase2a2* and *Ins2* promoters, detailed PCR parameters available upon request) or using a nested PCR protocol (*Sst* and *Glucagon* promoters). All PCR reactions were performed using HotStart Taq DNA polymerase (QIAGEN). For nested PCR: 95°C

**Table 1.** PCR primers for DNA methylation assays

| Genes | Round # | Primer sequences (5' to 3') (forward; reverse) |
| --- | --- | --- |
| *Amylase 2a* | Touch-down PCR | TTTTATTTTTATTTGGAATGGTG; TCATATTAAACCCAACAAAACC |
| *Insulin2* | Touch-down PCR | TTTAAGTGGGATATGGAAAGAGAGATA; ACTACAATTTCCAAACACTTCCCTAATA |
| *Glucagon* | Nested 1 | TTATATAATGTGGATGAGTGGG; TCTACCCTTCTACACCAAAATAC |
| *Glucagon* | Nested 2 | TTTGTTTGTTTAGATGAATGATT; TCTACCCTTCTACACCAAAATA |
| *Glucagon* | Nested 3 | AAGGGATAAGATTTTTAAATGAGA; TCTACCCTTCTACACCAAAATAC |
| *Glucagon* | Nested 4 | AAGGGATAAGATTTTTAAATGAGA; ACTCTCCAAACTATTTAACCTTACA |
| *Somatostatin* | Nested 1 | ATTGTTTGGTTTTTGTGGTATG; TCTTCCTTACCTCAAACAACC |
| *Somatostatin* | Nested 2 | TGGGTGTAGGTTTTTTTTTTTT; TCTTCCTTACCTCAAACAACC |

for 15 min followed by 45 cycles of 95°C/30 s, 52°C/30 s, 72°C/1 min, and last elongation at 72°C for 10 min. The final PCR product was purified using MinElute PCR Purification Kit (Qiagen), cloned, and sequenced. The sequences were analyzed using BiQ Analyzer software (Bock et al. 2005). For COBRA analysis (Combined bisulfite restriction analysis) of *Amylase 2a* promoter, the PCR product was generated and cut with Taqα1 enzyme. All primers used are listed in *Table 1*.

### *Ptf1aCreER* labeling of exocrine cells
*Ptf1aCreER;RosaYFP* double heterozygous animals were generated by mating homozygous *Cpa1CreER* males with *RosaYFP* homozygous females (Jackson lab). 2-month-old *Cpa1CreER;RosaYFP* adults were injected with Tamoxifen at 6 mg per animal every third day for four times to label acinar cells.

### Radioimmunoassay (RIA)
Exocrine tissues that contain the induced endocrine cells were harvested after pancreas dissociation and islets removal. After washing with Preculture Medium (1 and 5 mM glucose in PBS for Sst and Gcg RIA, respectively), exocrine tissue (approximately 20 mg per sample) and islets (20-30 islets per sample) were incubated in 1 ml pre-warmed Preculture Medium at 37° C for 1 hr. After removing Preculture Medium, the exocrine tissue and islet samples were re-suspended in 500 ml PBS in the absence or presence of 20 mM Arginine and incubated at 37° C for 1 hr. Supernatants were harvested. Somatostatin EURIA kit (Euro Diagnostica, Malmö, Sweden) and Glucagon RIA kit (Millipore, Darmstadt, Germany) were used to assess Somatostatin and Glucagon content of the incubation medium.

## Acknowledgements
We thank Andrew Kanarek and Lauren Grobicki for technical assistance; Dr Klaus Pechhold for assistance on intracellular FACS; Drs Kanako Miyabayashi, Kunio Kitamura, and Ken-ichirou Morohashi for the generous gift of Arx antiserum; Boston Children's Hospital core facility for Illumina array analysis; members of the Zhou lab for advice and feedback; and Juliana Brown and Jay Rajagopal for discussion and reading of the manuscript. QZ was supported by awards from the National Institute of Health. MN is supported by the Excellent Young Researcher Overseas visit Program from Japan Society for the Promotion of Science. WL is supported by a postdoctoral fellowship from the Juvenile Diabetes Research Foundation (JDRF). AZ is supported by postdoctoral fellowships from the Swiss Science Foundation (SNF) and the Swiss Foundation for Grants in Biology and Medicine (SFGBM).

## Additional information

### Funding

| Funder | Grant reference number | Author |
| --- | --- | --- |
| National Institutes of Health | 4 R00 DK077445 | Qiao Zhou |
| Juvenile Diabetes Research foundation | | Weida Li |

| Funder | Grant reference number | Author |
|---|---|---|
| Japanese Science Foundation | | Mio Nakanishi |
| Swiss Science Foundation | | Adrian Zumsteg |
| Japan Society for the Promotion of Science | | Mio Nakanishi |

The funders had no role in study design, data collection and interpretation, or the decision to submit the work for publication.

## Author contributions

WL, MN, QZ, Conception and design, Acquisition of data, Analysis and interpretation of data, Drafting or revising the article; AZ, MS, Acquisition of data, Analysis and interpretation of data, Drafting or revising the article; CW, Conception and design, Analysis and interpretation of data, Contributed unpublished essential data or reagents; DAM, Analysis and interpretation of data, Drafting or revising the article, Contributed unpublished essential data or reagents

## Ethics

Animal experimentation: This study was performed in strict accordance with the recommendations in the Guide for the Care and Use of Laboratory Animals of the National Institutes of Health. All of the animals were handled according to approved institutional animal care and use committee (IACUC) protocols (29-13) of Harvard University Faculty of Arts and Sciences. The Harvard University (HU)/Faculty of Arts and Science (FAS) animal-care and use program maintains full AAALAC accreditation, is assured with OLAW (A3593-01), and is currently registered with the USDA. All surgery was performed under approved and appropriate anesthesia, and every effort was made to minimize suffering.

# Additional files

## Major datasets

The following dataset was generated:

| Author(s) | Year | Dataset title | Dataset ID and/or URL | Database, license, and accessibility information |
|---|---|---|---|---|
| Li W, Zhou Q | 2013 | Induced pancreatic delta cells converted from acinar cells | GSE52522; http://www.ncbi.nlm.nih.gov/geo/query/acc.cgi?acc=GSE52522 | Publicly available at GEO (http://www.ncbi.nlm.nih.gov/geo/). |

The following previously published dataset was used:

| Author(s) | Year | Dataset title | Dataset ID and/or URL | Database, license, and accessibility information |
|---|---|---|---|---|
| Zhou Q | 2008 | Comparison of endocrine enriched genes in islet beta cells vs induced beta cells | GSE12025; http://www.ncbi.nlm.nih.gov/geo/query/acc.cgi?acc=GSE12025 | Publicly available at GEO (http://www.ncbi.nlm.nih.gov/geo/). |

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
