## [Decision Letter]

Thank you for sending your work entitled “In vivo reprogramming of pancreatic acinar cells to three islet endocrine subtypes” for consideration at *eLife*. Your article has been favorably evaluated by Senior editor, Janet Rossant, and 2 other reviewers.

The Senior editor and the other reviewers discussed their comments before we reached this decision, and the Senior editor has assembled the following comments to help you prepare a revised submission.

The paper of Li et al. is a sequel to an earlier publication in which the senior authors described the in vivo reprogramming of acinar cells into β islet cells in the pancreas, using a combination of three specific transcription factors. They now go an important step further, describing the effect of individual or dual combinations of these factors, which result in the formation of α- and δ-cells. They also deduce a sequence of synergistic and cross-antagonistic interactions of these factors that result in three defined cell states. The results are very interesting and overall convincing, although a few serious concerns remain.

Major concerns to be addressed:

1) The purification of endogenous and induced delta and alpha cells described in Figure 2—figure supplement 4 is unconvincing. First, since the reprogramming vectors introduced carry cherry as an infection marker it is unclear why the i-δ cells are cherry negative. Second, the i-α cells (D) could represent an autofluorescence artifact since a similar population can be seen in C. Here, a WT control must be shown, and signals recorded in irrelevant channels.

2) In Figure 2 the i-δ cells do not produce more somatostatin than islet control cells, although these contain only 5 % δ-cells (Figure 2—figure supplement 4). How can it then be concluded that 'the i-δ cells respond to arginine and released somatostatin, indicating that they possess the cellular machineries...' (penultimate paragraph in the Results section “Ngn3 converts acinar to δ-like cells”)? Minimally, to make this claim it is necessary to show the hormone production of i-δ cells not stimulated with arginine (as well as that of islet controls). Also, in the text it appears as if endogenous delta cells were tested, which was obviously not the case.

3) Please explain efficiency of co-infection. Assuming an infection rate of smaller than 100 % one would expect in a 3-factor infection using different viruses that there is at least a small fraction of single and double infected cells. It is therefore quite surprising why in the 3-factor condition not a single Sst or Gcg^+^ cell was found. Or similarly, Ngn3+Pdx1 co-infection should lead to some Ngn3-only infected cells. Have the authors validated with e.g., different fluorescence proteins that co-infection rates are indeed 100 %? Along the same lines, this requirement of 100 % infection efficiency to explain the data is somewhat at odds with the authors suggestion that low infection rates would explain their finding that with Ngn3+Mafa both Sst (presumably Ngn-only) and Gcg cells (presumably Ngn+Mafa).

4) How stable is the transformation long term, given that the cells are not in the right islet environment? They state δ- and α-cells are stable for at least 2 months – is this the end? Did the authors confirm that the exogenous transgenes are silenced? And in what time frame were they silenced?

5) The induced δ-cells could be sorted using mCherry from the vector – what time after induction was this – 10 days? How long does mCherry stay on in the induced cells? Can you use mCherry to sort the cells and do the in vitro secretion assays on sorted cells to ensure no contamination with exocrine cells?

6) Some discussion of the relevance of the findings to the regulatory pathways of normal pancreatic development would be helpful. The studies on using different vectors to assess the hierarchy of activation and repression of the different cell fates are nicely done and do provide a model of how this set of transcription factors can control cell fate conversion. What was not so clear was whether and how this regulatory hierarchy works during normal development of the pancreas. It is stated that the Ngn-induced state is not the same as that of an early pancreatic progenitor, but is Ngn normally involved in repressing acinar cell fate? More discussion on the similarities and differences between the induced state and the normal cell fate controls would be useful.

---

## [Author Response]

*1) The purification of endogenous and induced delta and alpha cells described in*
Figure 2—figure supplement 4
*is unconvincing. First, since the reprogramming vectors introduced carry cherry as an infection marker it is unclear why the i-δ cells are cherry negative. Second, the i-α cells (D) could represent an autofluorescence artifact since a similar population can be seen in C. Here, a WT control must be shown, and signals recorded in irrelevant channels*.

The reviewers are correct that both i-δ and i-α cells are cherry^+^. An observation we made that is relevant to the question raised by the reviewers is that upon fixation of cells with paraformaldehyde, Cherry fluorescence diminishes rapidly, indicating a quenching effect. Such a fixation step was performed in the intracellular FACS experiments and depending on the fixation time, the strength of the Cherry signal could vary significantly. A fixation time that differs by a few minutes made a difference in Cherry signal. In contrast, staining for hormones (somatostatin or glucagon) was performed after fixation and their fluorescence was unaffected by fixation length. Our records show that the original i-δ FACS data were collected from cells that went through longer fixation, thus the cherry fluorescence was weak. We now present FACS data with shorter fixation time, where stronger Cherry signals are apparent in i-δ cells (Figure 2—figure supplement 4).

On the second point raised by the reviewers, we now provide all the control FACS plots associated with i-α purification in Figure 3—figure supplement 2. These data clearly show that the signals are not autofluorescence.

*2) In*
Figure 2
*the i-δ cells do not produce more somatostatin than islet control cells, although these contain only 5 % δ-cells (*Figure 2—figure supplement 4*). How can it then be concluded that 'the i-δ cells respond to arginine and released somatostatin, indicating that they possess the cellular machineries...' (penultimate paragraph in the Results section “Ngn3 converts acinar to δ-like cells”)? Minimally, to make this claim it is necessary to show the hormone production of i-δ cells not stimulated with arginine (as well as that of islet controls). Also, in the text it appears as if endogenous delta cells were tested, which was obviously not the case*.

The original presentation of the data, as the reviewers pointed out, was unclear. We have now clarified this data set by showing the relative hormone secretion for i-δ and i-α (Figure 2 and Figure 3, respectively) with and without Arginine stimulation. We further added acinar cells as a control. Our data showed that acinar cells did not secrete somatostatin or glucagon in response to Arginine, whereas i-δ and i-α secreted hormone upon stimulation.

The reviewers are right that whole islets, not isolated endogenous cells, were used as controls. We thank the reviewers for pointing out this error. The text has now been corrected.

*3) Please explain efficiency of co-infection. Assuming an infection rate of smaller than 100 % one would expect in a 3-factor infection using different viruses that there is at least a small fraction of single and double infected cells. It is therefore quite surprising why in the 3-factor condition not a single Sst or Gcg*^*+*^
*cell was found. Or similarly, Ngn3+Pdx1 co-infection should lead to some Ngn3-only infected cells. Have the authors validated with e.g., different fluorescence proteins that co-infection rates are indeed 100 %? Along the same lines, this requirement of 100 % infection efficiency to explain the data is somewhat at odds with the authors suggestion that low infection rates would explain their finding that with Ngn3+Mafa both Sst (presumably Ngn-only) and Gcg cells (presumably Ngn+Mafa)*.

In this study, we used two different strategies to co-express two or three factors simultaneously. Where possible, our first choice is to use polycistronic co-expression. In this approach, multiple factors are expressed from a single construct in a single adenovirus. The co-expression is enabled by the viral 2A peptides placed between the factors. Published studies suggest that this polycistronic approach can lead to near 100 % co-expression of the factors (Szymczak and Vignali, 2005). Our studies also showed excellent co-localization of the 3-factors when expressed polycistronically (Figure 1—figure supplement 1). We observed that Polycistronic expression of the 3-factor indeed led to exclusive formation of i-β cells, with essentially no i-δ and i-alpha formation (after counting over 2,000 cherry^+^ cells, one or two Sst^+^ and glucagon^+^ cells were detected). Similarly, Ngn3.Pdx1 data were from polycistronic expression of Ngn3 and Pdx1 (Figure 1—figure supplement 2).

A second strategy we used for multi-factor expression is by co-injection of multiple adenoviruses with each carrying a single factor. The reviewers are right that co-expression by this method is not 100%. Our estimation is that it varies from 92-98% (49). We used co-injection mainly to induce i-α cells (Ngn3+Mafa) because this method yielded more i-α cells compared with polycistronic expression. The precise mechanism for this difference is unclear. We discussed these observations in the Results section and the Discussion section.

Getting back to the reviewers’ point, when we co-injected the 3-factor instead of polycistronic expression, we indeed observed small numbers of Sst^+^ and glucagon^+^ cells. Similarly, co-infection of Ngn3 and Pdx1 can lead to formation of a small number of Sst^+^ cells, as the reviewers predicted.

We have revised the figure legends and text throughout to clarify where polycistronic expression was used and where co-injection was used.

*4) How stable is the transformation long term, given that the cells are not in the right islet environment? They state δ- and α-cells are stable for at least 2 months – is this the end? Did the authors confirm that the exogenous transgenes are silenced? And in what time frame were they silenced*?

We have not tracked the induced cells beyond 2.5 months. We agree with the reviewers that outside the islet environment, the induced endocrine cells may not be able to persist long-term. Future experiments will be required to fully elucidate the long-term behavior of the induced cells.

We present new qPCR results and immunohistochemistry data to document the time line of transgene expression (Figure 1—figure supplement 3). qPCR results showed strong transgene expression 2 days after viral infection, substantial decline by day 10, and a return to baseline level by day 30. These results indicate that transgene expression mediated by adenovirus in adult mouse pancreas is transient, consistent with what we reported before (49).

*5) The induced δ-cells could be sorted using mCherry from the vector – what time after induction was this – 10 days? How long does mCherry stay on in the induced cells? Can you use mCherry to sort the cells and do the* in vitro *secretion assays on sorted cells to ensure no contamination with exocrine cells*?

The results presented in Figure 2 are from FACS isolated cherry^+^ cells 10 days after induction. Our observation from this study and others is that cherry expression can be detected in induced endocrine cells many months after the initial induction, although the fluorescence goes down with time. However, FACS significantly damages cells due to harsher enzymatic digestion required to dissociate tissue into single cells as well as the FACS procedure itself. Data from such damaged cells will not be reliable. The acinar cells do not express any of the endocrine hormones and they do not respond to Arginine (Figure 2 and Figure 3). Acinar cells present in the hormone secretion samples should therefore not interfere with the results.

*6) Some discussion of the relevance of the findings to the regulatory pathways of normal pancreatic development would be helpful. The studies on using different vectors to assess the hierarchy of activation and repression of the different cell fates are nicely done and do provide a model of how this set of transcription factors can control cell fate conversion. What was not so clear was whether and how this regulatory hierarchy works during normal development of the pancreas. It is stated that the Ngn-induced state is not the same as that of an early pancreatic progenitor, but is Ngn normally involved in repressing acinar cell fate? More discussion on the similarities and differences between the induced state and the normal cell fate controls would be useful*.

We thank the reviewers for this suggestion. We added a paragraph in the Discussion to discuss in length relevant information on the three factors in endocrine cell development during embryogenesis and compare their functions in reprogramming.